

# Assessment of a multi-resolution snow reanalysis framework: a multi-decadal reanalysis case over the Upper Yampa River Basin, Colorado.

Elisabeth Baldo[1] and Steven A. Margulis[1]

[1]Department of Civil and Environmental Engineering, University of California, Los Angeles, California, USA

*Correspondence to:* Elisabeth Baldo (ebaldo@g.ucla.edu)

**Abstract.** A multi-resolution (MR) approach was successfully implemented in the context of a data assimilation (DA) framework to efficiently estimate snow water equivalent (SWE) over a large head water catchment in the Colorado River Basin (CRB), while decreasing computational constraints by 60%. Thirty-one years of fractional snow cover area (fSCA) images derived from Landsat TM, ETM+ and OLI sensors measurements were assimilated to generate two SWE reanalysis datasets, a baseline case at a uniform 90 m spatial resolution and another using the MR approach. A comparison of the two showed negligible differences in terms of snow accumulation, melt and timing for the posterior estimates (in terms of both ensemble median and standard deviation). The MR approach underestimated the baseline peak SWE by less than 2%, and day of peak and duration of the accumulation season by a day on average. The largest differences were, by construct, limited primarily to areas of low complexity, where shallow snowpacks tend to exist. The MR approach should allow for more computationally efficient implementations of snow data assimilation applications over large-scale mountain ranges with accuracies similar to those that would be obtained using ~100 m simulations. Such uniform resolution applications are generally infeasible due to the computationally expensive nature of ensemble-based DA frameworks.

## 1 Introduction

Spatial resolutions of 100 m or less are more commonly being recommended when using land surface models (Wood et al. (2011), Bierkens et al. (2015), Beven et al. (2015)), especially when trying to capture the heterogeneity of snowpack states in montane regions (Clark et al. (2011), Winstral et al. (2014)). Previous work using hydrologic response units (HRUs; Beven and Kirby (1979), U. S. Geological Survey et al. (1983), Sivapalan et al. (1987), Chaney et al. (2016)), or triangulated irregular networks (TINs; Tucker et al. (2001), Vivoni et al. (2004), Mascaro et al. (2015)), showed that simulating in a "one size fits all" (uniform grid) approach is not only computationally expensive, but also sub-optimal since only small subsets of watersheds actually require being resolved at fine spatial resolutions. Along these lines, Baldo and Margulis (2017) developed a multi-resolution (MR) scheme for raster-based models and tested it in the context of deterministic snow modeling. By adapting the grid size to the physiographic complexity of the terrain, runtime and storage needs were cut in half while preserving the accuracy of a 90 m baseline simulation.

Deterministic forward modeling itself, even at high-resolution, is often insufficient due to errors in model inputs (most





notably precipitation) that are poorly characterized in montane regions. In lieu of deterministic modeling techniques, ensemble-based data assimilation (DA) methods are now frequently used to estimate snow states (Clark et al. (2006), Andreadis and Lettenmaier (2006), Su et al. (2008), De Lannoy et al. (2010), Liu et al. (2013), Arsenault et al. (2013), Girotto et al. (2014b), Margulis et al. (2015), Kumar et al. (2015)). The advantage of such approaches is to offer spatially and temporally continuous

estimates, while also providing a measure of their uncertainty. However, due to their ensemble nature, such methods can be extremely expensive to run at high spatial resolutions, which at least partly explains why many of the large-scale studies cited above simulate snow processes at resolutions on the order of 1 km or greater. Simulating at these scales can solve the computational issue, but inherently sacrifices valuable information related to sub-grid heterogeneities in montane regions. This is undesirable since relevant remote sensing data streams that can act as model constraints (e.g. Lidar, Landsat, MODIS, etc.)

are available at higher resolution (from meter- to hundreds of meter scale).

The recently developed 30+ year Sierra Nevada and Andes snow reanalysis datasets by Margulis et al. (2016) and Cortés and Margulis (2017) successfully leveraged high-resolution Landsat data using a data assimilation framework applied at uniform resolutions of 90 and 180 m respectively. For these regional-scale domains, this resulted in 6 million and 5.5 million simulation pixels respectively, which were run in the context of a 100-member ensemble. For reference, given that Northern Hemisphere

snow covered area is on the order of 8 million $km^2$ (Derksen and Brown (2012)), using a 100 m resolution would require the simulation of 8 billion pixels, a nearly four order of magnitude increase relative to the combined effort for the Sierra Nevada and Andes. Hence, extending these ensemble-based reanalysis methods to much larger scales using a uniform resolution on the order of 100 m is computationally prohibitive. Taking advantage of a MR approach to significantly reduce computational constraints might therefore greatly benefit ensemble-based DA frameworks and allow for applications at much larger scales.

This paper aims to test the performance of the MR approach from Baldo and Margulis (2017) in the context of a probabilistic DA framework (Margulis et al. (2015)).

The MR approach as applied by Baldo and Margulis (2017) only impacted prior (model-based) snow estimates as a result of aggregation of model inputs. In the context of the DA framework used by Margulis et al. (2016) and Cortés et al. (2016), the MR approach will also coarsen the fSCA observations derived from raw Landsat images (Cortés et al. (2014)), which can

potentially additionally impact the accuracy of the posterior snow state estimates. We hypothesize that this additional source of aggregation error will have minimal impact on the posterior estimates because it is expected a priori that the heterogeneity of fSCA in areas of low complexity will be minimal. Areas of high physiographic complexity typically correspond to areas of spatially heterogeneous snow accumulation and melt patterns, which drive fSCA evolution. Applying the MR approach to fSCA observations will therefore coarsen regions of the image where fSCA is most likely homogeneous and refine regions

where fSCA is most likely heterogeneous, and should therefore mitigate the impact of reducing the number of pixels on the reanalysis accuracy.

In this paper, a high-resolution (90 m) uniform grid baseline SWE reanalysis dataset was compared to one derived using the MR scheme to address the following questions: 1) How does the MR approach impact the assimilated fSCA observations? 2) How well does the MR approach perform in estimating the central tendency (i.e. ensemble median) of the posterior snow state

distribution in space and time? 3) How well does the MR approach perform in estimating the uncertainty of the posterior snow

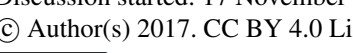



state distribution in space and time?

The rest of the paper is organized as follows: Section 2 illustrates the study area and the methodology used in this work. Section 3 compares the MR approach to the 90 m baseline case in order to answer the questions listed above. Finally, Section 4 summarizes the key points of this work.

## 2 Methodology

### 2.1 Study area

In order to maintain consistency with the work of Baldo and Margulis (2017), this study also used the Upper Yampa River Basin (UYRB, outlined in black in Figure 1) as a representative test domain of the Colorado River Basin (CRB). The CRB is large (6770 km$^2$) and snow-dominated, which makes it a critical source of fresh water for the 20 million people living downstream (Christensen et al. (2004)).

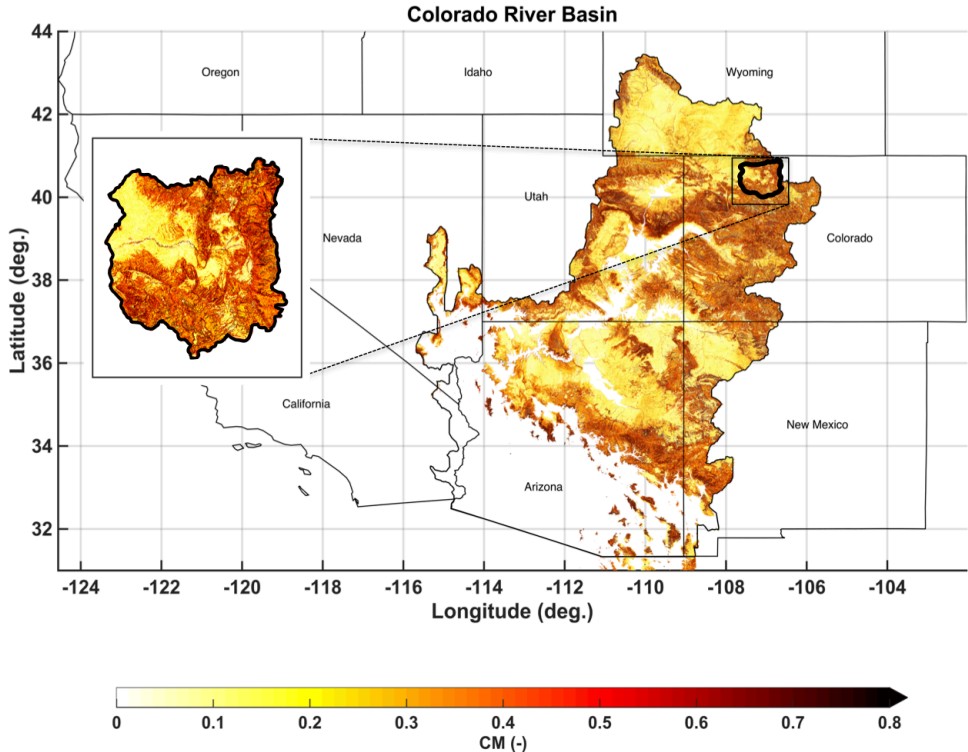

**Figure 1.** Complexity metric (*CM*) map of the Colorado River Basin (CRB) with the Upper Yampa River Basin (UYRB) outlined in black and displayed in more detail in the sub-panel.





In this study, the physiographic complexity metric (*CM*) was calculated for each 90 m pixel $i$ across the CRB (Figure 1) following the approach described in Baldo and Margulis (2017):

$$CM_i = \hat{\sigma_{Z_i}} + \hat{\sigma_{NI_i}} + \hat{\sigma_{fveg_i}} \tag{1}$$

where the normalized standard deviations of elevation ($\hat{\sigma_{Z_i}}$), and northness index ($\hat{\sigma_{NI_i}}$, Molotch et al. (2004)) were derived from the advanced spaceborne thermal emission and reflection (ASTER) global digital elevation model (DEM, JPL (2009)), and the normalized standard deviation of forested fraction ($\hat{\sigma_{fveg_i}}$) was derived from the National Land Cover Dataset (NLCD,

Homer et al. (2007)). Across the CRB, *CM* varies from 0 (bare and flat areas) to over 0.8 (steep and forested areas), with the UYRB sampling a similar range of complexity (Figure 1).

### 2.2 Multi-resolution approach

The MR algorithm begins with a pre-defined set of resolutions across which a raster-based model implementation will be applied. The finest baseline resolution is chosen to correspond to that deemed important for representing processes in high-

complexity areas of a basin. The specific set of resolutions to be applied are chosen by the user; herein we use factor 2 multiples of a 90 m baseline resolution up to 720 m. The final spatial distribution of resolutions depends on the choice of a maximum *CM* threshold ($CM_{max}$), above which pixels are simulated at the finest resolution and below which pixels are simulated at a mix of coarser resolutions. The threshold is chosen based on available computational resources for an application. In this study we chose to use a $CM_{max}$ of 0.65, which corresponds to the 90[th] percentile of the CRB *CM* values (Figure 2). Based on the

benchmarking tests performed by Baldo and Margulis (2017), such a threshold leads to a decrease in total pixel numbers on the order of 60 to 70%, which corresponds to reasonable computational costs for a full CRB snow reanalysis.

By construct, all of the UYRB pixels with a *CM* value larger than 0.65 were resolved at the baseline spatial resolution of 90 m, while the less complex ones were assigned either 720 m, 360 m, 180 m or 90 m by the MR algorithm developed by Baldo and Margulis (2017). The majority of the 720 m pixels are located in the northwestern part of the basin (Figure 2) corre-

sponding to flat and grassy areas. Modeling almost a quarter of the pixels at this coarse resolution represents the main source of computational savings, while minimizing the impact on snow accumulation and melt patterns given the homogeneous physiography of the terrain. The remaining low *CM* pixels were assigned either 360 m, or 180 m depending on the complexity of their neighbors. In terms of the most complex pixels, 31% of the pixels are resolved at 90 m in order to preserve the accuracy of SWE estimates. In UYRB, these pixels tend to be located at higher elevations, where the terrain is rugged and densely forested

as described in Baldo and Margulis (2017) (Figure 2).



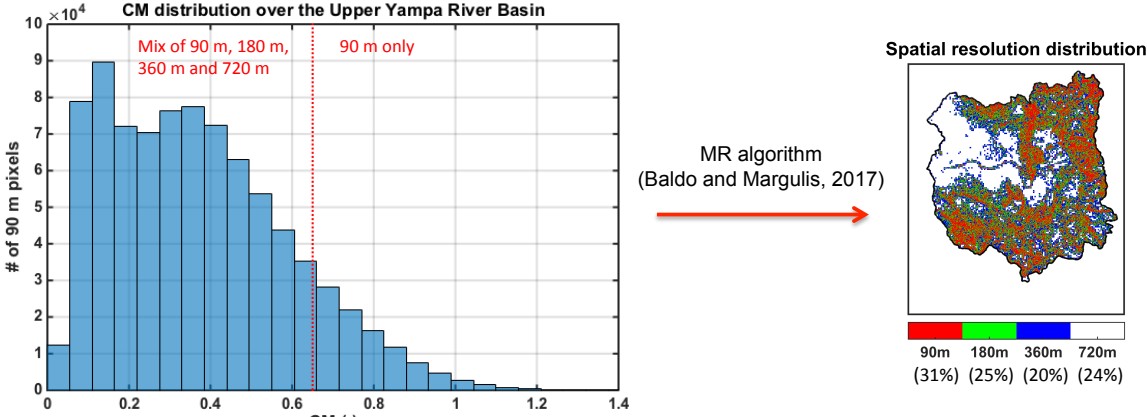

**Figure 2.** (left panel) Complexity metric distribution for the Upper Yampa River Basin. The choice of the maximum threshold $CM_{max}$ of 0.65 represented as the red vertical line leads to (right panel) the spatial resolution distribution map.

## 2.3 SWE reanalysis framework

### 2.3.1 Model framework and forcings

The modeling setup used in this study is the same as described in Margulis et al. (2016). The Simplified Simple Biosphere (SSiB) model developed by Xue et al. (1991), coupled with a three layer snow and atmosphere soil transfer (SAST) model (Sun and Xue (2001), Xue et al. (2003)) was used as the land surface model (LSM) to represent the interactions between the atmosphere, vegetation, and snow. A snow depletion curve (SDC) (Liston (2004)) was used to represent the sub-grid heterogeneity in SWE and the resulting fSCA. The coupled LSM-SDC generates time series of SWE and fSCA as a function of the sub-grid coefficient of variation (CV) and pixel-averaged cumulative snowfall and snowmelt.

The static inputs required by the LSM are latitude, longitude, elevation, slope, and aspect, which were derived from the advanced spaceborne thermal emission and reflection (ASTER) DEM (JPL (2009)), as well as landcover derived from the National Land Cover Database (NLCD, Homer et al. (2007)). The static inputs were aggregated from their original 30 m resolution to the model resolution (either 90 m for the baseline or a mix of 90 m, 180 m, 360 m, and 720 m for the MR case). The dynamic meteorological forcings were obtained from the Phase 2 North American Land Data Assimilation System (NLDAS-2, Cosgrove et al. (2003), Xia et al. (2012)) hourly forcing dataset. NLDAS-2 variables include precipitation, incident shortwave radiation, near-surface air temperature, humidity, wind speed and pressure at a coarse spatial resolution of 1/8 °. The NLDAS-2 forcings were downscaled to the model resolution using topographic correction methods that have been previously applied over the Sierra Nevada and the Andes (Girotto et al. (2014b), Girotto et al. (2014a), Margulis et al. (2016) and Cortés et al. (2016)) as well as Upper Yampa in Baldo and Margulis (2017). Lapse rates of 6.5°K/km and 4.1°K/km were used for air temperature and dewpoint temperature respectively. Downscaling approaches for atmospheric pressure, specific humidity, and the incoming longwave and shortwave radiation fluxes are explained in detail in Girotto et al. (2014b) (Appendix A). The downscaling is





not deterministic, but also incorporates a priori uncertainty in the forcings (Girotto et al. (2014b); Appendix A). It is important to note that the precipitation is not downscaled a priori, but treated as an uncertain random variable following a lognormal distribution with a mean of 2.25 and a standard deviation of 0.5 that is then implicitly downscaled and updated as part of the data assimilation framework.

### 2.3.2 Assimilation of Landsat-based fractional snow cover area using a particle batch smoother

The probabilistic DA framework used in this study is referred to as the Particle Batch Smoother, or PBS, and was developed by Margulis et al. (2015) in order to improve the probabilistic reanalysis framework used previously for SWE reanalysis in Durand et al. (2008), Girotto et al. (2014b) and Girotto et al. (2014a). The coupled LSM-SDC provides a prior ensemble esti-

mate for all snow states and fluxes based on the specified input uncertainty and its propagation through the model. The prior ensemble treats each replicate as an equally likely (equal weight) realization based on the postulated input uncertainty. The goal of the PBS approach is to optimally weight the different uncertainty sources coming from the meteorological forcing and fSCA retrievals in order to generate posterior snow estimates. Specifically, the reanalysis step is applied to a batch of the full set of fSCA measurements (retrospectively) over the water year. A likelihood function updates the prior weights whereby the

posterior weights can be used to determine the pdf or moments (i.e. mean, median, variance, inter-quartile range, etc.) of any of the snow states/fluxes. The mathematical framework is presented in detail in Margulis et al. (2015).

Landsat-5 thematic mapper (TM), Landsat-7 enhanced thematic mapper (ETM+), and Landsat-8 operational land imager (OLI) images from water year 1985 to 2015 were used to calculate fSCA and fractional vegetation cover over each pixel. For a given sensor, measurements are available every 16 days at a spatial resolution of 30 m, and only clear-sky images were

processed to obtain fSCA. The raw data consist of multispectral top of atmosphere radiance measurements that are transformed into top of atmosphere reflectance before being atmospherically corrected. The spectral unmixing algorithm validated by Cortés et al. (2014) and based on Painter et al. (2009) then retrieves the fraction and type of constituent (snow, vegetation or bare rock/soil) within each pixel through a least-square-error optimization. The linear unmixing model estimates reflectances from each constituent and selects the combination of constituents leading to the lowest root mean square error (RMSE) between

the modeled reflectance and a library of snow reflectances that have previously been calculated for different combinations of constituents within each pixel. The validation of the algorithm by Cortés et al. (2014) showed an fSCA retrieval error of approximately 15%. The vegetation cover fraction (fVEG) was also retrieved from the spectral unmixing algorithm and annually averaged and used within the LSM-SDC. The fVEG derived from Landsat observations was chosen over the static NLCD for use in the LSM-SDC model to allow for inter-annual variability and because it is also, by construct, more consistent with the

fSCA observations used in the assimilation step. Similar to the static input data, the fSCA and fVEG images at 30 m were then aggregated to either 90 m for the baseline case, or a mix of 90 m, 180 m, 360 m, and 720 m for the MR case.




### 2.3.3 Verification of posterior SWE estimates

A posterior set of SWE reanalysis estimates was first generated for 31 years (WY 1985 – WY 2015) at the baseline resolution of 90 m, and compared to in-situ measurements to assess its accuracy. A total of 203 peak SWE measurements from six SNOTEL stations and 1421 monthly manually sampled SWE from seven snow courses were used. Not all locations have full records for

the full period, with two snow pillows / courses starting in 1986 and one in 1998. All snow pillows are collocated with snow courses and station 5 is a snow course only (Figure 3). All in-situ observations are taken at high elevations, between 2500 and 3200 m, in densely forested clearings; some representativeness errors are therefore expected when compared to grid-averaged SWE estimates.

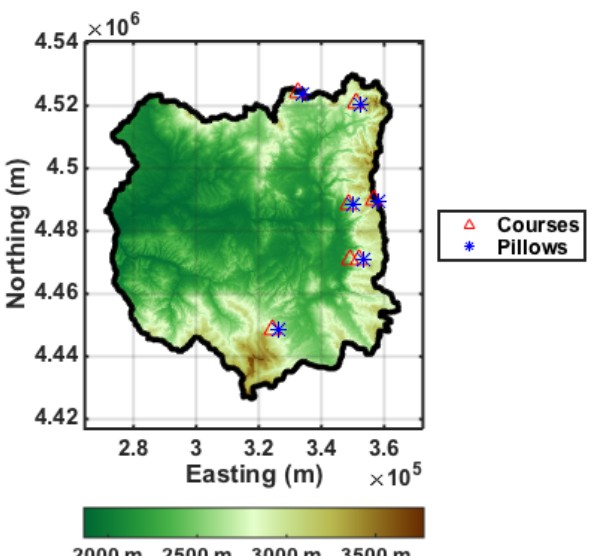

**Figure 3.** Elevation map of the Upper Yampa River Basin with the location of the seven snow courses shown in red and the location of the six snow pillows shown in blue.

The prior SWE estimates are highly uncertain by construct, and overestimated in–situ observations from both snow courses and pillow (Figure 4). Prior estimates had a mean difference (MD) of 30 cm, and a root mean square difference (RMSD) of 41 cm for snow courses, and a MD of 43 cm, with a RMSD of 51 cm for snow pillows. Both showed a similar correlation coefficient ($R^2$) of 0.86. Note that, based on previous work (Luo et al. (2003), Girotto et al. (2014b)), the NLDAS-2 precipitation was assumed biased and therefore bias-corrected using the prior distribution (using a mean of 2.25 as indicated above). The

fact that the prior SWE overestimates in situ data is an indication that there is likely an over-correction in the prior precipitation (at least at these sites). In contrast, the reanalysis generated posterior SWE estimates that are much more consistent with the in-situ data, are extremely well correlated to in-situ measurement and show limited mean differences. The MD is less than 2 cm




for snow courses and less than 5 cm for snow pillows, with RMSD of 10 cm and $R^2$ higher than 0.95 for both. The small differences observed may be partly explained by undercatch problems with SNOTEL pillows measurements, and also by the fact that in-situ SWE measurements are usually made in easily accessible areas such as clearings and therefore not fully representative of the collocated 90 m pixel-average values. The difference in errors between the prior and posterior is primarily indicative of

5   the data assimilation method properly selecting ensemble members with precipitation forcing that is consistent with the fSCA observations. Based on the comparison with in situ data, the posterior SWE estimates generated at 90 m can be considered to be an accurate representation of the true underlying SWE for the UYRB and are thus used as a baseline throughout.

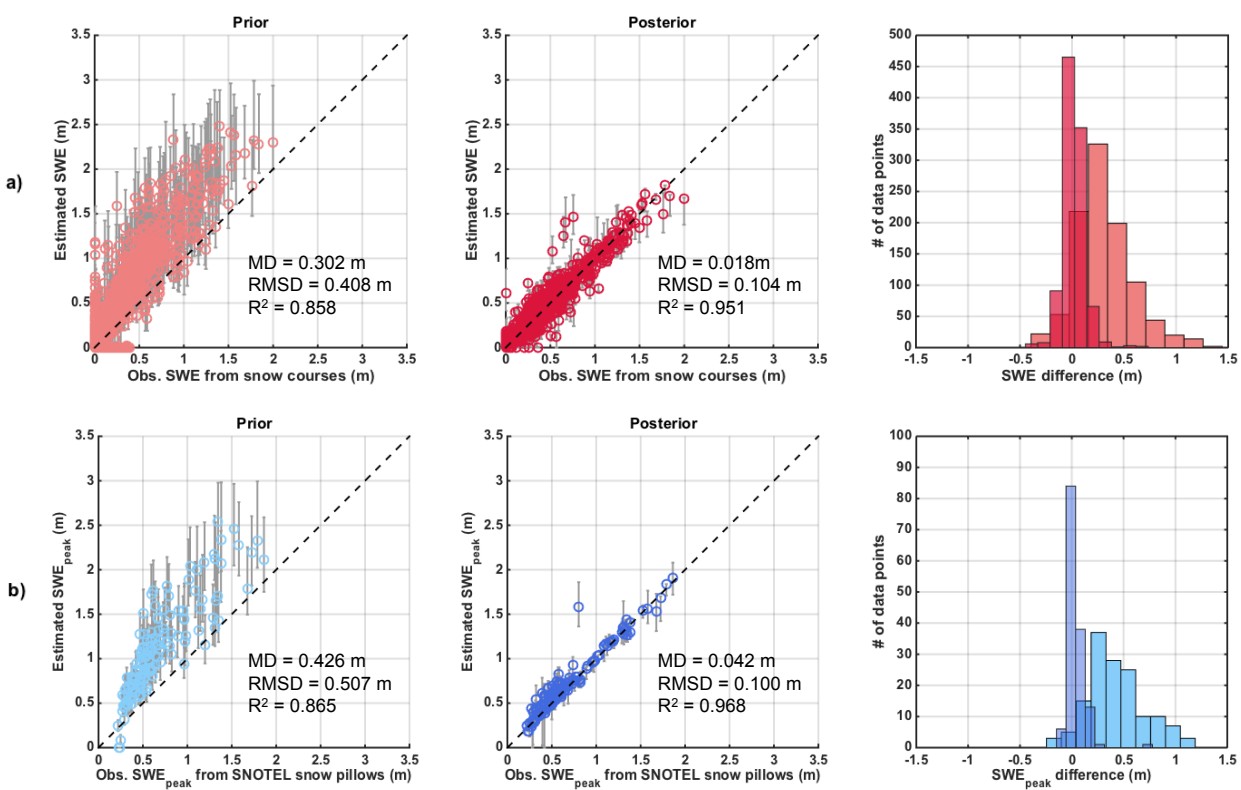

**Figure 4.** (left panel) Scatter plots of prior estimated snow water equivalent (SWE) vs. in situ, (middle panel) posterior estimated SWE vs. in situ and (right panel) histogram of the difference for (a) all snow courses and (b) snow pillows. The markers represent ensemble medians while the intervals represent the interquartile range (IQR). The mean difference (MD), root mean square difference (RMSD) and correlation coefficient ($R^2$) are displayed.



## 3 Performance of the MR SWE reanalysis compared to the 90 m baseline

As shown previously in section 2.3.3, performing a SWE reanalysis at 90 m yields an accurate reference solution for our test basin. However, such a simulation is very expensive in terms of computational resources. Modeling the basin uniformly at 90 m meant running almost 840,000 pixels with an ensemble size of 100 replicates, which took over a month on the UCLA

computer cluster and required 850G of space to store the resulting outputs. On the other hand, the MR approach decreased the number of pixels and storage need by 59%. Since pixels are simulated independently from each other, they are run in parallel, which is why runtime also decreased by 59% and took less than 2 weeks. Knowing that the MR SWE reanalysis can decrease computational constraints by a factor of two or more, the following section aims to assess its performance in terms of accuracy.

### 3.1 Impact of the MR approach on the assimilated fSCA observations

The MR modeling approach as applied previously in Baldo and Margulis (2017) impacts the prior snow simulations, but in the context of a DA (reanalysis) framework as done herein, it also coarsens the fSCA observations that provide the key constraint that generates the posterior estimates. Assessing the difference between the baseline and MR fSCA is therefore crucial to understand the full effect of the MR approach on the data assimilation step. To this end, processed fSCA images at the 90 m baseline and at the MR were first compared during the accumulation season, around day of peak (*DOP*) and during the ablation

season. As seen in Figure 5, the MR approach does not significantly alter the fSCA observations for the three sample dates chosen, and the mean absolute difference (MAD) is on the order of 5% (the MD is 0% for all three measurements) over the UYRB. The largest differences are concentrated over areas with partial snow cover (notably the Southeastern corner in Figure 5a and the more central parts of the basin in Figure 5b-c), which most likely correspond to snow ablation.



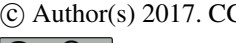

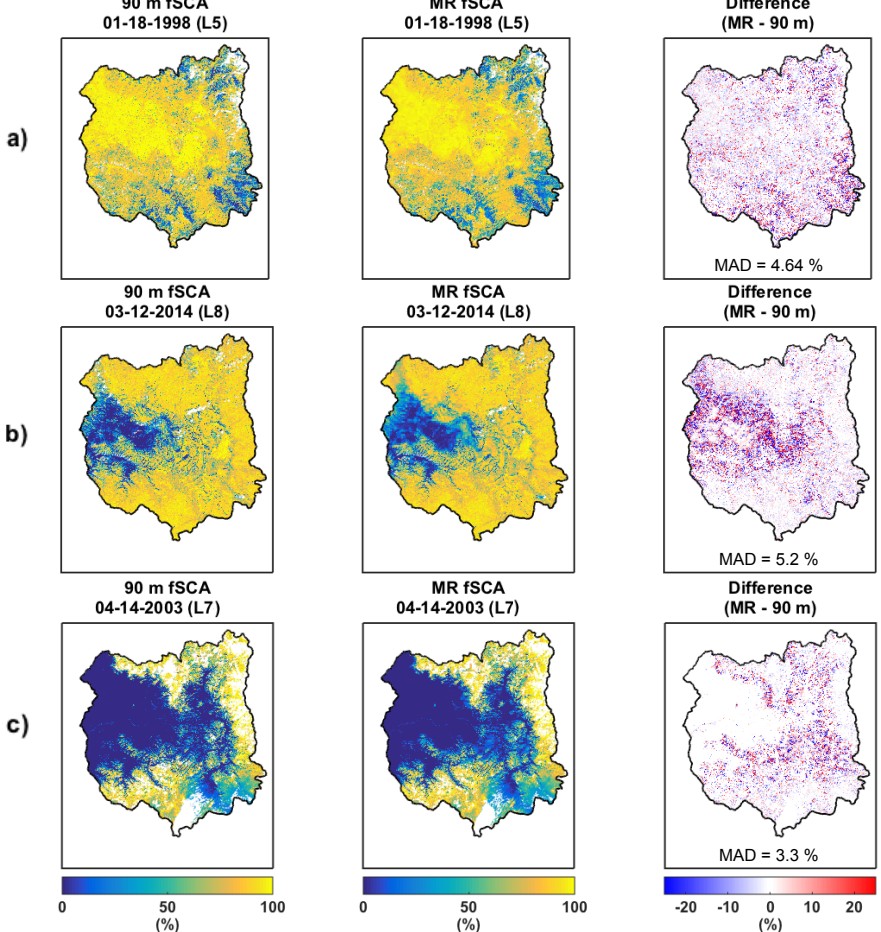

**Figure 5.** Maps of fractional snow cover area (fSCA) during (a) the accumulation season, (b) at a time near day of peak (*DOP*), and (c) during the ablation season over the Upper Yampa River Basin for the 90 m baseline, the multi-resolution (MR) case and the difference between the two approaches. The exact acquisition day and sensor type (L5 for Landsat-5 TM, L7 for Landat-7 ETM+, and L8 for Landat-8 OLI) are displayed for all three samples. White areas inside the watershed bounds (in the left and middle panels) were covered by clouds.

In order to better understand the seasonality of the fSCA differences, all observations were binned by month and averaged over the 31 years of record (Figure 6a). The differences are negligible between the 90 m baseline and the MR case during the accumulation season (October to January), while the MR method slightly overestimates the baseline fSCA by 4% or less during the ablation season (February to August). The annual average difference is 0.87%. The expected impact of assimilating larger fSCA values during the ablation season is an overestimation of the length of the snowmelt period, which, for the same amount of melt season energy inputs, would translate into larger posterior SWE estimates. As seen in Figure 6b-c, fSCA from both the baseline and the MR case share a similar distribution with respect to *CM* and Peak SWE ($SWE_{peak}$). As expected,



areas of high fSCA correspond to areas of high SWE accumulation at the higher elevation of the basin, which also tend to be the most complex. By design, the MR approach does not coarsen areas of high physiographic complexity that can experience sharp differences in accumulation/ablation from one pixel to another. Hence, by construct, the MR fSCA is identical to the baseline for CM larger than 0.65, and slightly differs from the baseline in low complexity areas as seen in Figure 6b. In addition, Figure 6c shows that the difference in fSCA over regions of high SWE accumulation is negligible as well (1.3% or less). Given the small differences observed, the effect of the MR approach on the assimilated fSCA observations is minimal and therefore is not expected to significantly alter the performance of the data assimilation scheme (discussed in more detail below).

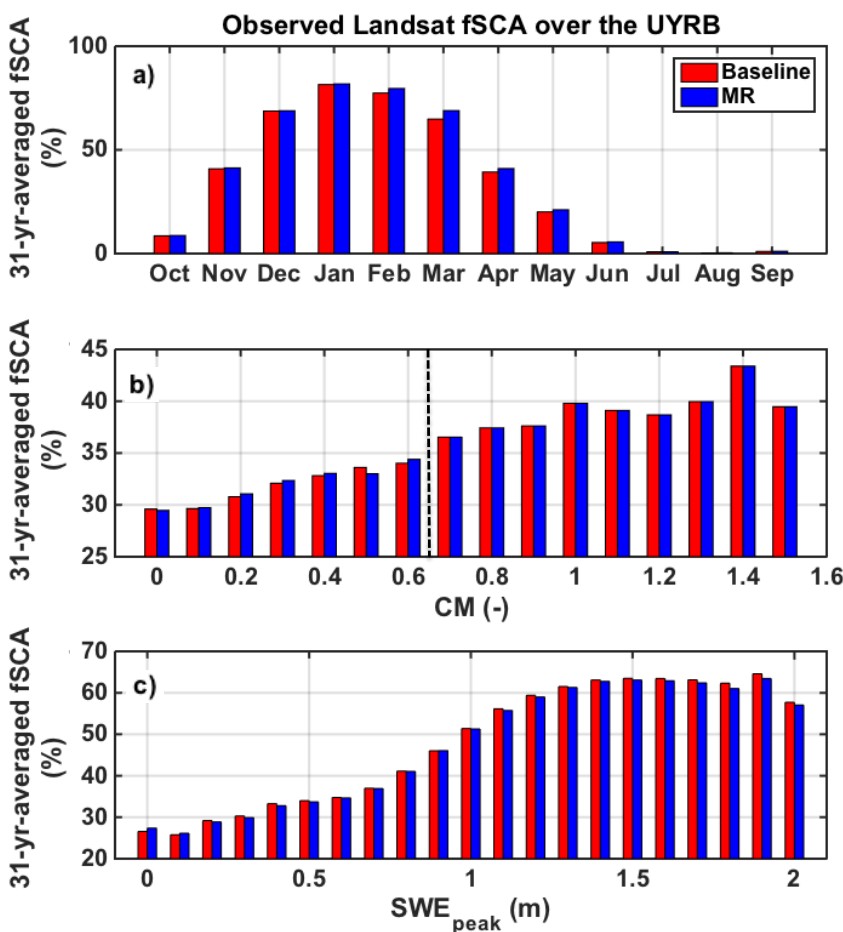

**Figure 6.** fSCA climatology derived from the 31-yr record of Landsat observations over the Upper Yampa River Basin: bin-averaging of all observations across the range of (a) months of the water year, (b) $CM$ values, and (c) peak SWE ($SWE_{peak}$) values for the 90 m baseline and MR case. The $CM$ maximum threshold $CM_{max}$ of 0.65 is represented by the vertical dashed line (b).





## 3.2 Impact of the MR approach on snow climatology metrics

The following analysis focuses on the comparison of the posterior ensemble median SWE estimates for the baseline and MR cases. Peak SWE ($SWE_{peak}$), day of peak ($DOP$), and duration of melt ($DOM$) were chosen for analysis. $SWE_{peak}$ is defined as the maximum daily SWE in a given WY. $DOP$ is defined for each WY as the day when SWE is equal to $SWE_{peak}$. $DOM$ is

5  the difference between the melt-out day, defined as the day when only 1% of the original $SWE_{peak}$ remains, and $DOP$, which effectively quantifies the duration of the ablation season. These metrics can be defined either pixel-wise or for basin-averaged values.

### 3.2.1 Mean spatial distribution

10  Figures 7a, 8a, and 9a show maps of the 31-yr average pixel-wise $SWE_{peak}$, $DOP$ and $DOM$, while figures 7b, 8b, and 9b show the distribution of the respective 31-yr average relative differences binned by *CM*, elevation (*Z*), slope, fVEG, and $SWE_{peak}$. In these figures, the baseline estimates were always subtracted from the MR estimates, which means that a positive difference represents an overestimation of the baseline by the MR case and vice versa.

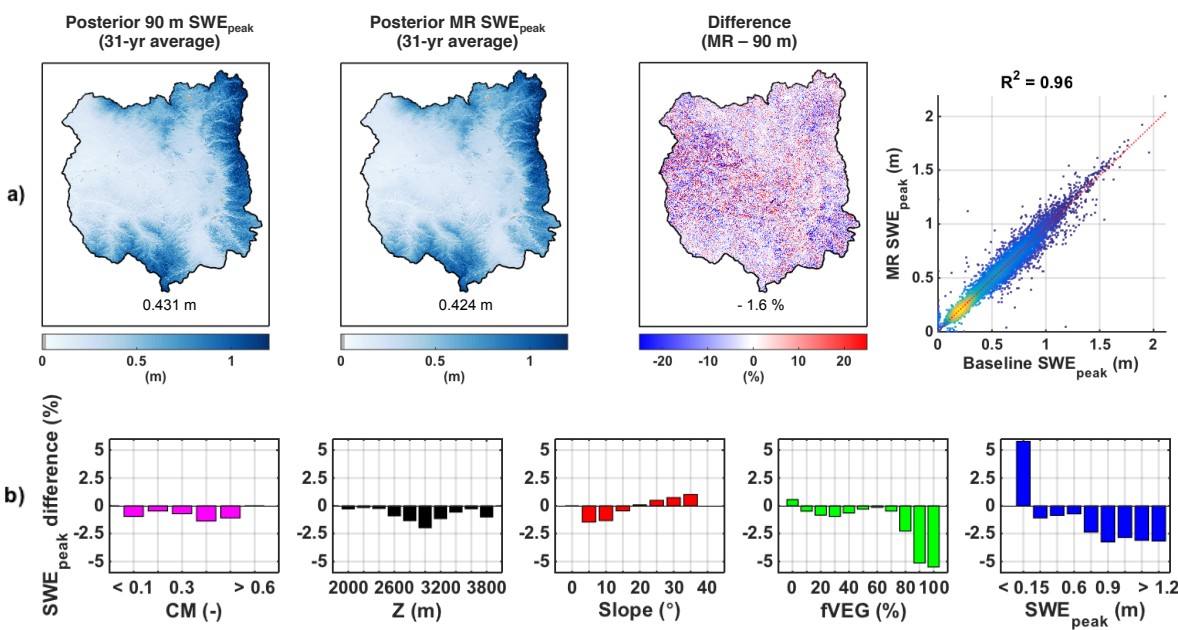

**Figure 7.** (a) Maps of pixel-wise 31-year average posterior peak SWE ($SWE_{peak}$) over the Upper Yampa River Basin for the 90 m baseline, the MR case, the percent difference between the two approaches (MR – baseline), and the corresponding scatter plot. Basin averages are displayed at the bottom of each map. (b) Distribution of $SWE_{peak}$ relative difference with complexity metric (*CM*), elevation (*Z*), slope, forested fraction (fVEG), and $SWE_{peak}$. Pixels with a 31-yr average $SWE_{peak}$ lower than 5 cm were discarded from the analysis.





As expected, the climatological $SWE_{peak}$ shows significant spatial variability for both the MR and 90 m baseline with values ranging from zero to well over 1 m of SWE (Figure 7a). The middle and western parts of the basin that are not physiographically complex (see Figure 1) receive 25 cm or less on average. Given their location and relatively low elevation (less than 2000 m) the SWE accumulation is not orographically driven, but more heavily influenced by the few winter snowstorms occurring over

the basin. The more complex areas in the eastern and southern edges of the basin accumulate a much larger amount of SWE (on the order of 1 m or more). On average, the MR approach underestimated pixel-wise $SWE_{peak}$ by 7.2 mm or 1.6%, with the most complex areas showing no difference since they were modeled at 90 m by design, and the less complex but high elevation areas showing larger differences on the order of 10 cm, or roughly 10% of $SWE_{peak}$. As seen in the density scatter plot, the majority of pixels have a $SWE_{peak}$ around 20 cm, and the correlation between the baseline and the MR case is very

strong with a correlation coefficient of 0.96. Figure 7b shows that the bin-averaged relative differences between the pixel-wise MR and baseline $SWE_{peak}$ are constrained between -5 and 5%. By construct, the CM bands larger than 0.65 show no difference because all the MR pixels were simulated at the baseline resolution. All elevations bands show an underestimation of $SWE_{peak}$, with the largest differences observed at middle elevations between 2600 m and 3200 m. Since the UYRB is densely forested at these elevations, this is consistent with the largest underestimation occurring for the highest fVEG bands. Regarding

the distribution of the differences with slope, the lower slope bands ($0° - 15°$) underestimate $SWE_{peak}$ while the higher slope bands ($20° - 35°$) show overestimation. As discussed in Baldo and Margulis (2017), the coarsening of pixel properties by the MR method leads to a slight increase in fVEG for densely vegetated pixels, as well as an increase of more gentle sloped and north facing pixels. In the context of the SWE reanalysis, the magnitude of melt energy flux largely dictates the peak SWE that is consistent with a given fSCA depletion time series. The increase in fVEG as a result of the MR approach leads to an

underestimation of the melt (energy) flux at the snow surface (as a result of attenuation of solar radiation), which decreases the posterior MR $SWE_{peak}$ for these pixels. Since the minimum solar zenith angle during the ablation season over the UYRB is 16°, reducing gentle slopes ($0° - 15°$) leads to an underestimation of the melt flux (as a result of becoming less perpendicular to the incoming direct beam solar radiation), which decreases the posterior MR $SWE_{peak}$ for these pixels. Reducing steeper slopes ($20° - 35°$) has the opposite effect and overestimates the melt flux, increasing the posterior MR $SWE_{peak}$ for these

pixels.

The posterior $SWE_{peak}$ estimates are therefore impacted by the MR approach in two ways: i) an overestimation of the assimilated fSCA during the ablation season and ii) a general underestimation of the melt flux due to the coarsening of the basin physiography, with the exception of steep pixels where the melt flux is overestimated. The basin-averaged underestimation of $SWE_{peak}$ observed in Figure 7 suggests that the effect of coarsening the static inputs and meteorological forcing on $SWE_{peak}$

is more important than the effect from the coarsened assimilated fSCA images. More importantly, the differences are the largest for the lowest $SWE_{peak}$ band (less than 15 cm). The MR approach therefore concentrated the largest $SWE_{peak}$ differences to areas of low *CM* that tend to accumulate less SWE.





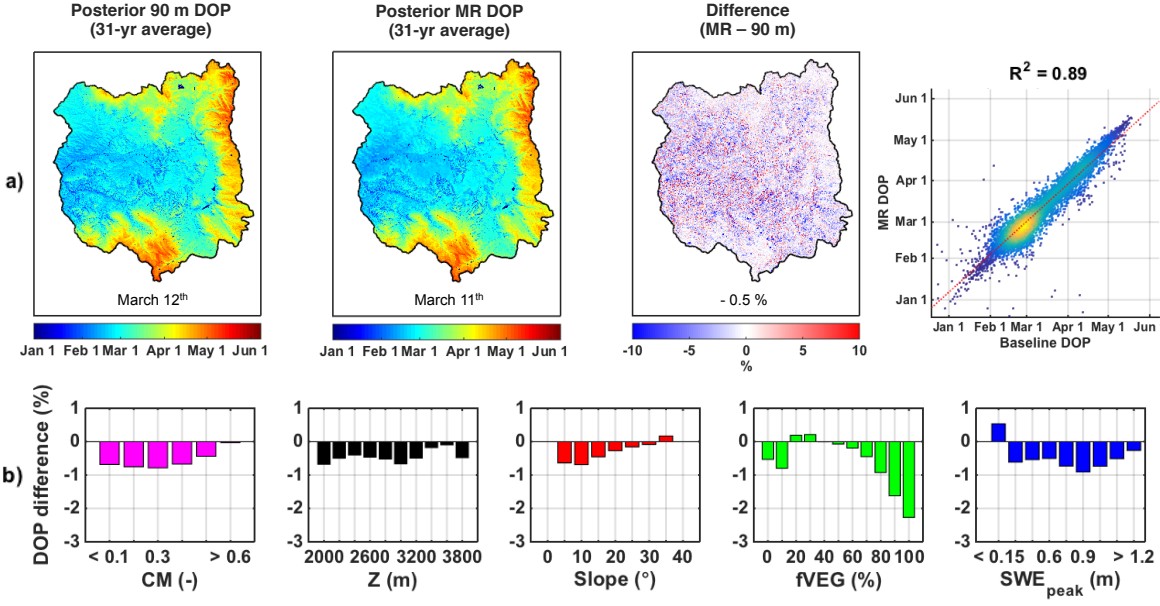

**Figure 8.** (a) Maps of pixel-wise 31-year average day of peak (*DOP*) over the Upper Yampa River Basin for the 90 m baseline, the MR case, the percent difference between the two approaches (MR – baseline), and the corresponding scatter plot. Basin averages are displayed at the bottom of each map. (b) Distribution of *DOP* relative difference with complexity metric (*CM*), elevation (*Z*), slope, forested fraction (fVEG), and $SWE_{peak}$. Pixels with a 31-yr average $SWE_{peak}$ lower than 5 cm were discarded from the analysis.

Regarding *DOP*, Figure 8a shows that SWE in the middle and western regions of the basin that are not physiographically complex peaks early during the winter between January and March. In contrast, the more complex regions in the eastern and southern parts of the UYRB accumulated SWE until much later during the spring (April to June). These complex regions show very good agreement between the baseline and MR case in term of timing, with larger differences over the rest of the basin.

5 The average underestimation of 0.8 day or - 0.5% is negligible. As seen in the density scatter plot, the majority of pixels have peak values around March 1st, with a strong correlation coefficient of 0.89. Figure 8b shows *DOP* difference distributions with *CM*, elevation, slope, fVEG and $SWE_{peak}$ similar to $SWE_{peak}$ (Figure 7b), while the magnitude of the *DOP* differences is much smaller and ranges between 0.5% and -2%. The MR approach therefore preserves the accuracy of areas accumulating large amounts of SWE, that peak later in the spring.





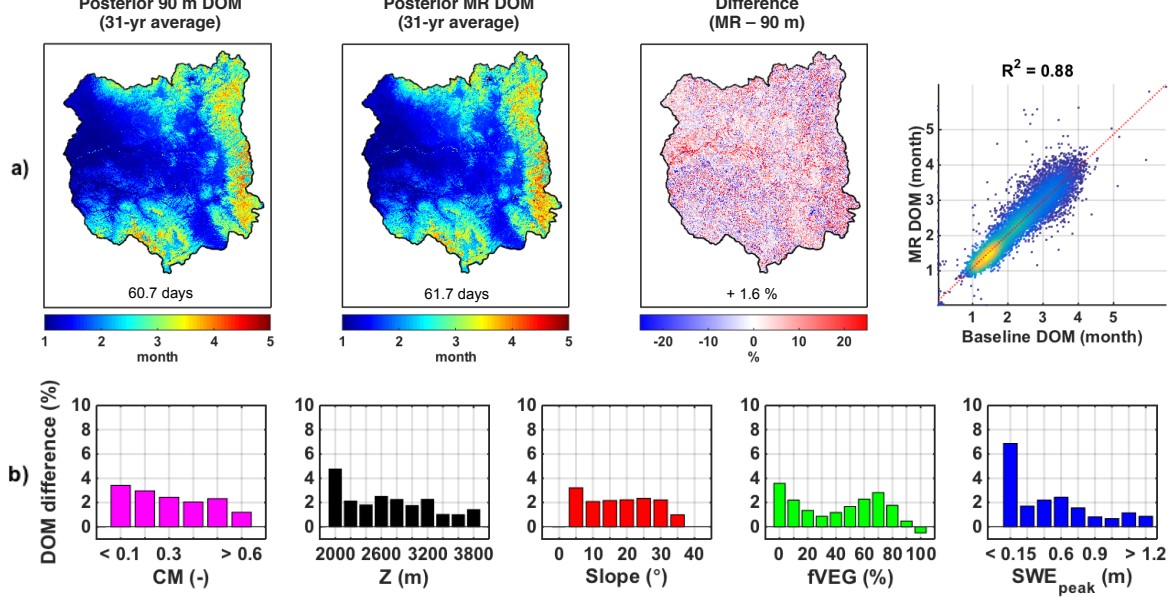

**Figure 9.** (a) Maps of pixel-wise 31-year average duration of melt (*DOM*) over the Upper Yampa River Basin for the 90 m baseline, the MR case, the percent difference between the two approaches (MR – baseline), and the corresponding scatter plot. Basin averages are displayed at the bottom of each map. (b) Distribution of *DOM* relative difference with complexity metric (*CM*), elevation (*Z*), slope, forested fraction (fVEG), and $SWE_{peak}$. Pixels with a 31-yr average $SWE_{peak}$ lower than 5 cm were discarded from the analysis.

Regarding the duration of the ablation season, *DOM* can vary from less than a month over the areas that accumulated little SWE and started melting as soon as the snowstorm events ended, to almost five months over the southwestern edge of the basin (Figure 9a). The average *DOM* is 61.7 days, or 2 months for the MR case, which overestimates the 90 m baseline by 1 day or 1.6%. The density scatter plot shows that the majority of pixels have a *DOM* between one and two months, with a strong

5 correlation coefficient of 0.88. The slight overestimation of *DOM* by the MR case was expected, given the underestimation of melt fluxes from the increase of gentle north facing and densely forested pixels (Baldo and Margulis (2017)) and the higher assimilated fSCA observations in the MR case. Figure 9b shows that the largest *DOM* overestimation occurs at the lowest band for all five variables. When looking at the distribution with $SWE_{peak}$ specifically, pixels accumulating 15 cm of SWE of less show a *DOM* difference of 7%, while pixels accumulating 1 m or more only show a *DOM* difference of 1% or less. Pixels

10 accumulating low amounts of SWE can be very intermittent in nature, without a clear $SWE_{peak}$ or *DOP*, which can explain the higher difference seen in Figure 9b.

Based on these results, when applying the MR approach to the SWE reanalysis framework, we therefore expect the largest differences to occur over areas of low physiographic complexity. These types of areas tend to peak early during the winter,





accumulate less SWE, and melt within a month and display lower levels of spatial variability that are easier to model at coarser resolutions.

### 3.2.2 Basin-average mean seasonal cycle

The mean seasonal cycle of MR SWE underestimates the baseline case by less than 1 cm as shown by the 31-yr average difference displayed in black in Figure 10b. Figure 10a-b show that the seasonal cycles for both the MR and baseline case closely match during the accumulation season (November to March) with differences in the range of +1 / -1 cm and a negative mean around – 5 mm as shown by the grey shaded area and the black line respectively in Figure 10b. The basin-averaged mean $SWE_{peak}$ is 0.374 m for the MR case, and 0.381 m for the 90 m baseline (Figure 10a), which leads to a mean difference of -7.1 mm (or -1.95%, Figure 10b). In terms of timing, $DOP$ based on the mean seasonal cycle fits almost perfectly within a day, with the MR case peaking on March 15[th] and the baseline case on March 16[th] on average (Figure 10a). The underestimation is more pronounced during the early ablation season (March to June), where the difference in assimilated fSCA observations is the largest (Figure 6a) with the entire range of WYs showing negative differences, and a maximum of -2.1 cm (or -5.4%) observed for the wettest year, WY 1996. Even though the MR case is assimilating slightly larger fSCA observations during the ablation season (Figure 6a), the coarsening of the static inputs shown in Baldo and Margulis (2017) decreases the energy inputs, which ultimately lowers the posterior MR SWE estimates, and therefore explains the slight underestimation observed during the ablation season in Figure 10.





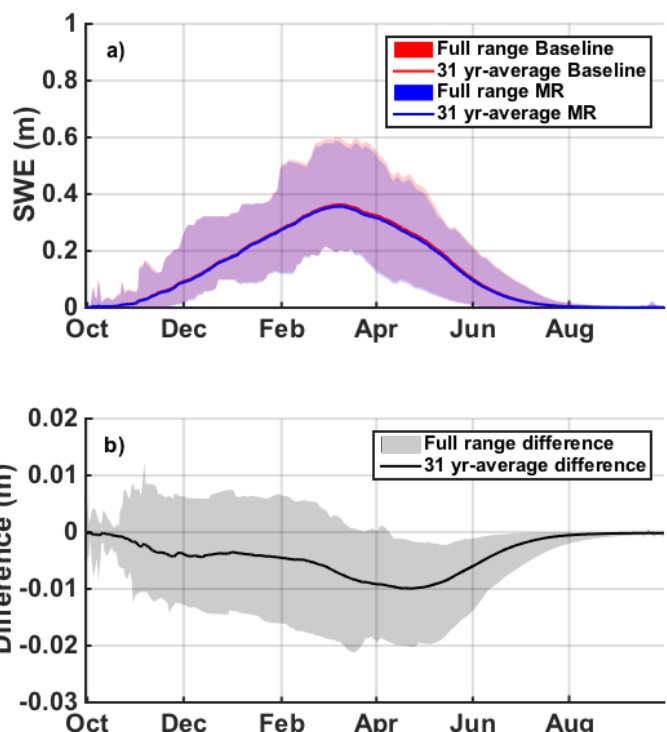

**Figure 10.** (a) Daily timeseries of basin-averaged posterior SWE from WY 1985 to WY 2015. The 31-yr averages are displayed in solid lines, while the shaded regions represent the full range across WYs. (b) The 31-yr averaged difference between the MR case and the baseline is displayed in black, with the full range of differences shaded in grey.

### 3.2.3 Inter-annual variability

The baseline and MR annual timeseries of $SWE_{peak}$ show close agreement in inter-annual variations (Figure 11a). The scatter plot illustrates the positive performance of the MR case, including at both ends of the spectrum, which confirms that the MR case is estimating dry and wet years accurately. Figure 11b-c also illustrates the similarities in *DOP* and *DOM* inter-annual variability. WY 1985 shows the largest differences because there were two similar values of maximum SWE within 1 cm that occurred 15 days apart. The MR case identified the first peak as $SWE_{peak}$, while the baseline did the opposite, which does not impact the $SWE_{peak}$, estimate, but does impact both *DOP* and *DOM*. Beyond this single-year, the MR case closely represents the inter-annual variability in the timing and length of accumulation and ablation seasons over the reanalysis period.



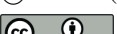

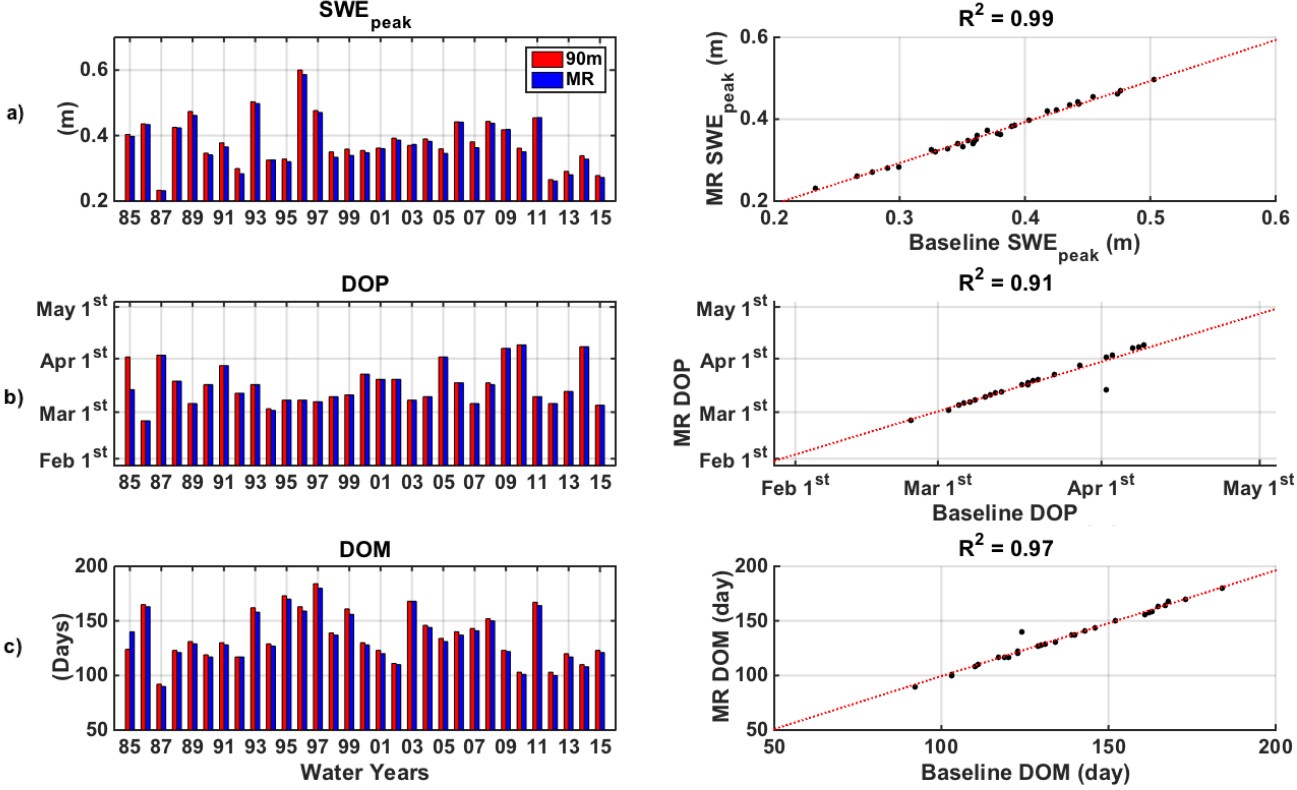

**Figure 11.** (left panel) Annual timeseries and (right panel) scatter plot with linear regressions of basin-averaged (a) peak SWE ($SWE_{peak}$), (b) day of peak ($DOP$), and (c) duration of melt ($DOM$) or the 90 m baseline and the MR case.

The fact that the MR case is capturing inter-annual variability correctly is further confirmed by the similar shapes of the baseline and MR empirical distribution functions (ECDFs) for all three metrics (Figure 12). The 10th percentiles, medians and 90th percentiles for both the MR and baseline cases are presented in Table 1. The statistics between the baseline and MR cases only differ by 1 cm for $SWE_{peak}$, and 2 days for *DOM*. Regarding *DOP*, the statistics are identical (Table 1), and the

5 difference between the two ECDFs in Figure 12b is due to the special conditions in WY 1985 as explained above. Half of the WYs had $SWE_{peak}$ less than 0.37 m, while 10% had less than 0.29 m and 90% had less than 0.47 m for the baseline, or 0.36 m, 0.28 m, and 0.46 m for the MR case. The *DOP* distribution was identical between the baseline and MR cases, with the median, 10th and 90th percentiles on March 12, March 4 and April 2 respectively. Finally, *DOM* was shorter than 303 days for the baseline or 301 days for the MR case for half of the WYs simulated, with the 10th and 90th percentiles being 278 days / 276

10 days and 327 days / 325 days respectively.





| Percentile | Baseline | | | Multi-Resolution (MR) | | |
|---|---|---|---|---|---|---|
| | $SWE_{peak}$ (m) | $DOP$ | $DOP$ (day) | $SWE_{peak}$ (m) | $DOP$ | $DOP$ (day) |
| 10$^{th}$ | 0.29 | March 4 | 278 | 0.28 | March 4 | 276 |
| 50$^{th}$ | 0.37 | March 12 | 303 | 0.36 | March 12 | 301 |
| 90$^{th}$ | 0.47 | April 2 | 327 | 0.46 | April 2 | 325 |

**Table 1.** Return period values for peak SWE ($SWE_{peak}$), day of peak ($DOP$), and duration of melt ($DOM$) for both the baseline and MR cases.

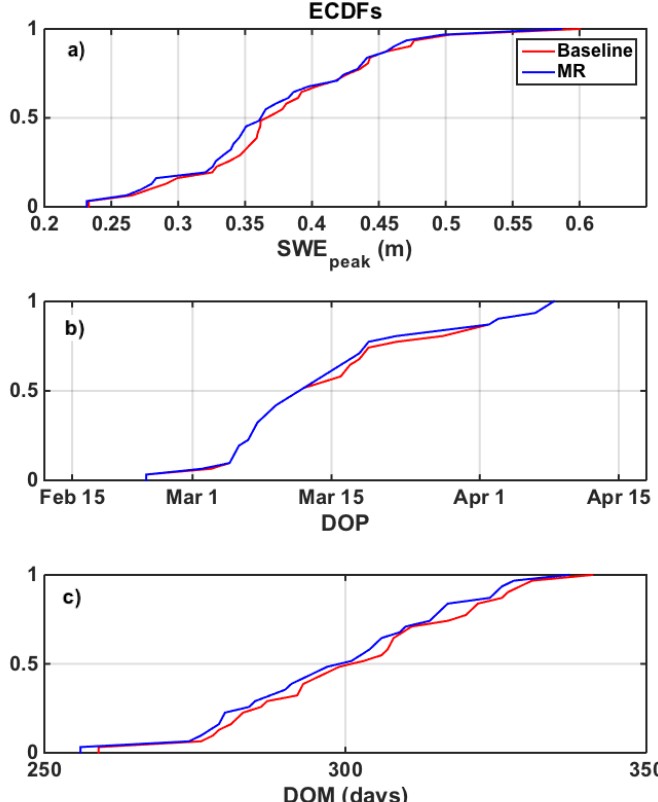

**Figure 12.** Empirical cumulative distribution functions (ECDFs) of (a) peak SWE ($SWE_{peak}$), (b) day of peak ($DOP$), and (c) duration of melt ($DOM$) for the 90 m baseline and the MR case.





### 3.3 Impact of the MR approach on spatial variations of SWE uncertainty

The previous analysis focused on the impact of the MR approach on the posterior ensemble SWE median (i.e. a metric of central tendency). However, another strength of the reanalysis framework is to also provide a measure of uncertainty via the posterior ensemble. In this section the impact of the MR approach on the posterior ensemble $SWE_{peak}$ standard deviation ($<\sigma>$) and coefficient of variation ($<CV>$) is examined, where the angle brackets ($< >$) are used to emphasize the ensemble operator.

In order to focus on the spatial distribution of the ensemble posterior $SWE_{peak}$ uncertainty, the 31-yr average maps of $<\sigma>$ and $<CV>$ (Figure 13) were created by pooling $<\sigma>$ and $<CV>$ for each pixel over all 31 WYs as follows (Bingham and Fry (2010)):

$$\overline{<\sigma>_i} = \sqrt{\frac{\sum_{y=1}^{31}(<\sigma>_i^y)^2 + \sum_{y=1}^{31}(<\mu>_i^y - \overline{<\mu>_i})^2}{31}}$$

$$\overline{<CV>_i} = \frac{\overline{<\sigma>_i}}{\overline{<\mu>_i}} \tag{2}$$

where the overbar notation denotes the 31-year average. $\overline{<\sigma>_i}$ is the 31-yr average ensemble $SWE_{peak}$ standard deviation for pixel $i$, and $\overline{<\mu>_i}$ is the 31-year average ensemble $SWE_{peak}$ mean for the same pixel $i$. $<\sigma>_i^y$ and $<\mu>_i^y$ are respectively the ensemble $SWE_{peak}$ standard deviation and mean for each individual WY $y$. The 31-yr average $SWE_{peak}$ coefficient of variation ($\overline{<CV>_i}$) for each pixel $i$ was calculated as the ratio between the pixel 31-yr average ensemble $SWE_{peak}$ standard deviation and mean.



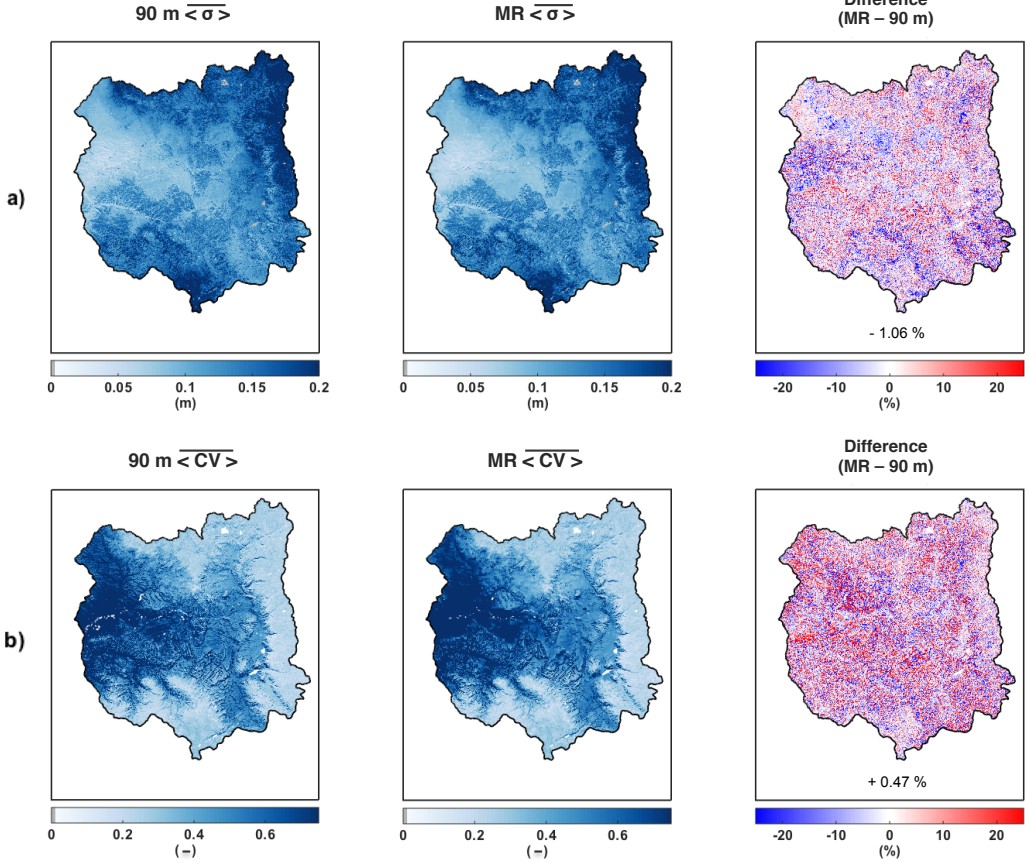

**Figure 13.** Maps of the 31-yr average ensemble $SWE_{peak}$ (a) standard deviation ($\overline{<\sigma>_i}$), and (b) coefficient of variation ($\overline{<CV>_i}$) for (left panel) the 90 m baseline, and (middle panel) MR case. The relative differences in $\overline{<\sigma>_i}$ and $\overline{<CV>_i}$ are shown in the right panel. The basin-averaged differences are displayed at the bottom of the maps in the right panel.

As seen in Figure 13, the spatial distributions of $\overline{<\sigma>}$ and $\overline{<CV>}$ are highly variable. For both the baseline and MR cases, the high elevation areas accumulating large amounts of SWE (see Figure 7a) show $\overline{<\sigma>}$ on the order of 15 - 20 cm with a $\overline{<CV>}$ on the order of 10% – 20%, while the lower parts of the UYRB have a $\overline{<\sigma>}$ around 5 cm or less, with a $\overline{<CV>}$ higher than 60%. Regarding the relative difference between the MR and baseline cases (Figure 13a-b, right panel), no particular spatial pattern can be observed for $\overline{<\sigma>}$, with the exception of a few areas showing an underestimation on the order of 10%, bringing the basin-average difference to -1.06% or -1.8 cm. Figure 13b shows that the regions accumulating the most SWE with the lowest $\overline{<CV>}$ also have the lowest relative difference between the MR and baseline cases (white areas on the eastern and southern edges of the UYRB). Similar to the difference in $\overline{<\sigma>}$, the basin-average difference in $\overline{<CV>}$ of 0.47% is negligible.





## 4 Conclusions

This study demonstrated the performance of a new MR terrain discretization approach in the context of a snow reanalysis framework using the assimilation of Landsat-derived fSCA observations. The MR approach was shown to have an insignificant impact on the fSCA observations assimilated and the reanalysis framework led to posterior SWE ensembles similar to the high-resolution 90 m baseline. The SWE reanalysis dataset generated with the MR approach matched the 90 m baseline ensemble median within 1 cm on average for peak SWE magnitude and within 1 day on average for timing of the accumulation and melt seasons. Most of the difference between the two approaches occurs in areas accumulating less than 15 cm of SWE, while areas accumulating more than that are estimated with a high degree of accuracy. In addition, the MR approach also preserved the SWE uncertainty, where the ensemble standard deviation and coefficient of variation showed differences on the order of -1% and 0.5% respectively. This study has demonstrated the feasibility of the MR approach in the context of a snow reanalysis framework, where the significant decrease in computational costs will allow much larger scale implementations of the SWE reanalysis over full mountain ranges, while preserving the accuracy of fine spatial resolution simulations.

*Competing interests.* The authors declare that they have no conflict of interest.

*Acknowledgements.* This work was partially supported by a NASA NEWS project (grant NNX15AD16G) and the National Science Foundation (grant number EAR-1246473). The raw data presented herein include the ASTER DEM (available at http://asterweb.jpl.nasa.gov/), the National Land Cover Database (NLCD; available at http://www.mrlc.gov/), the NASA NLDAS-2 forcing data set (available at http://ldas.gsfc.nasa.gov/nldas/), and the Landsat reflectance data (available at http://earthexplorer.usgs.gov/). All simulations were performed by using computational and storage services associated with the Hoffman2 Shared Cluster provided by the UCLA Institute for Digital Research and Education's Research Technology Group.





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
