# Peer review of "Assessment of a multi-resolution snow reanalysis framework: a multi-decadal reanalysis case over the Upper Yampa River Basin, Colorado."

_Hydrology and Earth System Sciences, 2017_

## Referee Comment (RC1) · Anonymous Referee #1 · 3 Jan 2018

General comments:

This study applies the multi-resolution (MR) grid discretization approach developed by Baldo and Margulis (2017) to a data assimilation test case. Fractional snow covered area (fSCA) is assimilated in a snow reanalysis system using a particle batch smoother for a portion of the Yampa River in Colorado, USA. The MR approach uses concepts of hydrologic similarity as justification to aggregate portions of the landscape into coarser resolution units, thus saving computational time with little loss of information. Baldo and Margulis (2017) show good results in the initial testing and thus apply

the MR approach to an assimilation case. The MR approach is able to recreate the high-resolution simulation with minimal differences. The authors do a reasonable job explaining the deviations between the two simulations.

Overall, the paper is very straightforward to understand, is generally well written, and reasonably organized. This is an incremental study of the MR approach and similarity concepts more generally. However, it is sufficient for publication in that it does demonstrate clearly that the MR approach can be used successfully with fSCA assimilation at a reduced computational load, which is a clean result for future data assimilation studies.

Specific comments:

1) Are there any other observations during the reanalysis period that can be used to look at the spatial distribution of SWE across the basin? Overflights from the Airborne Snow Observatory (ASO), or any measurements from the NASA CLPX perhaps? This is a curiosity comment as it isn't critical for the paper.

2) It would be nice to recreate the density scatter plot in Figure 7a for all 31-years of peak SWE, and discuss any outliers that may be found. Mean analysis will mask any year-to-year performance differences, which may provide deeper insight into this use of the MR approach for assimilation. Also, outlier years and corresponding performance of model estimates in those years are key for water resource managers.

This would complement or be added to the basin average yearly analysis in section 3.2.3.

3) Same comment as 2) for Figures 8 and 9.

4) For Figure 13 it would be nice to see the distribution of the differences as well.

---

## Referee Comment (RC2) · Anonymous Referee #2 · 10 Jan 2018

In the manuscript "Assessment of a multi-resolution snow reanalysis framework: a multi-decadal reanalysis case over the Upper Yampa River Basin, Colorado", the authors present a multi-resolution approach for a snow data assimilation application in the mountainous region of the Colorado River Basin. They assimilate fSCA estimates from different satellite-based remote sensing systems and LSM results to generate a SWE reanalysis dataset. They present a comparison of prior and posterior (to the assimilation of remote sensed data) SWE results with station measurements as a validation of the assimilation framework. In the results section, the performance of the MR approach

is discussed in detail. Main aim of the MR approach is reducing the computing time for large scale applications and datasets.

General Comments

The authors present a comprehensive application of a data assimilation and MR framework. They present state-of-the-art data and approaches and apply their own data assimilation and MR framework to a large test case. The approach to compare the MR case with a baseline resolution case in the results section is well chosen and nicely elaborated. However, the authors could think of shortening the results section at some points, because the results are somehow quite obvious from the beginning of the results section on. The differences in posterior SWE when coarsening the resolution for less complex terrain in the MR method compared to the baseline case are very small or even negligible. This is an impressive result and underlines clearly the validity to use the MR approach when a reduction of CPU and storage resources is needed. However, as said, the results section could be much shorter, because the differences are so small that most of the figures present a more or less "perfect match". Besides showing the performance of the MR method, the 31-year dataset gives very interesting insights in the snow climatology of the region. The manuscript is technically of very high quality and well written, and I recommend its publication after minor revisions.

Specific Comments

L. 15: Rephrase and remove "by the user": e.g. "Factor 2 multiples . . . are chosen in this study as specific set of resolutions."

---

## Author Comment (AC1) · 9 Feb 2018

1) Are there any other observations during the reanalysis period that can be used to look at the spatial distribution of SWE across the basin? Overflights from the Airborne Snow Observatory (ASO), or any measurements from the NASA CLPX perhaps? This is a curiosity comment as it isn't critical for the paper:

ASO data is unfortunately not yet available over the Upper Yampa basin. CLPX is, but was not included in the verification because the focus of this work was to compare

posterior MR SWE to the baseline. The verification performed using SNOTEL data was not meant to be exhaustive, but rather provide confidence that the baseline was reasonable. Other datasets can later be included for verification when the method is applied over larger domains.

2) It would be nice to recreate the density scatter plot in Figure 7a for all 31-years of peak SWE, and discuss any outliers that may be found. Mean analysis will mask any year-to-year performance differences, which may provide deeper insight into this use of the MR approach for assimilation. Also, outlier years and corresponding performance of model estimates in those years are key for water resource managers. This would complement or be added to the basin average yearly analysis in section 3.2.3. Same comment as 2) for Figures 8 and 9:

Based on the organization of the paper, Section 3.2.2 intentionally focused on the mean annual patterns. We therefore would argue that adding inter-annual variability in that section may hurt the flow of the paper. The interannual performance was presented in the next two sections, but focused on basin-averaged metrics, because extracting information from the scatterplots using all 31 years metrics presented was thought to be too noisy. An example of the scatter plots including all 31 years of data are shown below in Figure 1. These results are consistent with the mean annual plots presented in the paper (but noisier as expected). Given the lack of additional information brought by the scatter plots in Figure 1, as well as the existing thorough inter-annual analysis already presented, we feel that including these three new figures does not add significant insight and therefore would plan to exclude them from the result section of the revised manuscript.

3) For Figure 13 it would be nice to see the distribution of the differences as well:

We welcome the suggestion, and implemented the scatter plot and difference distribution of the ensemble uncertainty coefficient of variation (CV) in Figure 13 to mimic what is done for Figures 7-9. Since another reviewer suggested to shorten the result
[Figure]

section, we are planning on focusing on CV only in the revised manuscript, instead of both CV and standard deviation as currently shown. As seen in Figure 2, the differences between the MR and baseline SWEpeak CVs remain within the same range as for the ensemble median (5-10%). Further, differences are mostly constrained over low complexity and SWEpeak areas, which supports the conclusion that the MR approach also preserves the accuracy of the ensemble uncertainty over areas of importance for montane snow processes.

[Figure]

[Figure]

**Fig. 1.** Density scatter plots of baseline vs. MR (left) SWEpeak, (middle) DOP, and (right) DOM for all 31 years, as well as their respective linear regression plot. The linear regression coefficient is presen

[Figure]

**Fig. 2.** (a) Maps and scatter plot of pixel-wise 31-year average posterior peak SWE (SWE-peak) coefficient of variations and (b) Distribution of SWEpeak coefficient of variation relative difference

---

## Author Comment (AC2) · 9 Feb 2018

1) "the authors could think of shortening the results section at some points, because the results are somehow quite obvious from the beginning of the results section on."

We welcome this suggestion to shorten the result section, and will do so with an aim for improving its flow in the revised manuscript.

2) L. 15: Rephrase and remove "by the user": e.g. "Factor 2 multiples . . . are chosen in this study as specific set of resolutions."

[Figure]

We agree with the reviewer, and will modify the revised manuscript accordingly.

---

## Author Response (AR1)

**Response to Reviewers**

**"Assessment of a multi-resolution snow reanalysis framework: a multi-decadal reanalysis case over the Upper Yampa River Basin, Colorado" by Baldo and Margulis.**

The reviewers' comments are reproduced below in black and our corresponding responses are presented in blue. In the responses, any references to page numbers, sections, or figures correspond to the revised manuscript.

**Reviewer #1 Comments:**
General comments: This study applies the multi-resolution (MR) grid discretization approach developed by Baldo and Margulis (2017) to a data assimilation test case. Fractional snow-covered area (fSCA) is assimilated in a snow reanalysis system using a particle batch smoother for a portion of the Yampa River in Colorado, USA. The MR approach uses concepts of hydrologic similarity as justification to aggregate portions of the landscape into coarser resolution units, thus saving computational time with little loss of information. Baldo and Margulis (2017) show good results in the initial testing and thus apply the MR approach to an assimilation case. The MR approach is able to recreate the high-resolution simulation with minimal differences. The authors do a reasonable job explaining the deviations between the two simulations. Overall, the paper is very straightforward to understand, is generally well written, and reasonably organized. This is an incremental study of the MR approach and similarity concepts more generally. However, it is sufficient for publication in that it does demonstrate clearly that the MR approach can be used successfully with fSCA assimilation at a reduced computational load, which is a clean result for future data assimilation studies.

1) Are there any other observations during the reanalysis period that can be used to look at the spatial distribution of SWE across the basin? Overflights from the Airborne Snow Observatory (ASO), or any measurements from the NASA CLPX perhaps? This is a curiosity comment as it isn't critical for the paper.

ASO data is unfortunately not yet available over the Upper Yampa basin. CLPX is, but was not included in the verification because the focus of this work was to compare posterior MR SWE to the baseline. The verification performed using SNOTEL data was not meant to be exhaustive, but rather provide confidence that the baseline was reasonable. Other datasets can later be included for verification when the method is applied over larger domains.

2) It would be nice to recreate the density scatter plot in Figure 7a for all 31-years of peak SWE, and discuss any outliers that may be found. Mean analysis will mask any year-to-year performance differences, which may provide deeper insight into this use of the MR approach for assimilation. Also, outlier years and corresponding performance of model estimates in those years are key for water resource managers. This would complement or be added to the basin average yearly analysis in section 3.2.3. 3) Same comment as 2) for Figures 8 and 9.

Based on the organization of the paper, Section 3.2.2 intentionally focused on the mean annual patterns. We therefore would argue that adding inter-annual variability in that section may hurt the flow of the paper. The interannual performance was presented in the next two sections, but

focused on basin-averaged metrics, because extracting information from the scatterplots using all 31 years metrics presented was thought to be too noisy. An example of the scatter plots including all 31 years of data are shown below in Figure A. These results are consistent with the mean annual plots presented in the paper (but noisier as expected). Given the lack of additional information brought by the scatter plots in Figure A, as well as the existing thorough inter-annual analysis already presented, we feel that including these three new figures does not add significant insight and therefore would plan to exclude them from the result section of the revised manuscript.

[Figure]

**Figure A.** Density scatter plots of baseline vs. MR (left) $SWE_{peak}$, (middle) $DOP$, and (right) $DOM$ for all 31 years, as well as their respective linear regression plot. The linear regression coefficient is presented in each panel title.

4) For Figure 13 it would be nice to see the distribution of the differences as well.

We welcome the suggestion, and implemented the scatter plot and difference distribution of the ensemble uncertainty coefficient of variation (CV) in Figure 11 in the revised manuscript to mimic what is done for Figures 6-8. Since another reviewer suggested to shorten the result section, we are planning on focusing on CV only in the revised manuscript, instead of both CV and standard deviation as initially shown. As seen in Figure B, the differences between the MR and baseline SWEpeak CVs remain within the same range as for the ensemble median (5-10%). Further, differences are mostly constrained over low complexity and SWEpeak areas, which supports the conclusion that the MR approach also preserves the accuracy of the ensemble uncertainty over areas of importance for montane snow processes.

[Figure]

**Figure B.** (a) Maps of pixel-wise 31-year average posterior peak SWE ($SWE_{peak}$) coefficient of variation over the Upper Yampa River Basin for the 90 m baseline, the MR case, the percent difference between the two approaches (MR – baseline), and the corresponding scatter plot. (b) Distribution of $SWE_{peak}$ coefficient of variation relative difference with complexity metric ($CM$), elevation ($Z$), slope, forested fraction (fVEG), and $SWE_{peak}$. Pixels with a 31-yr average $SWE_{peak}$ lower than 5 cm were discarded from the analysis.

**Reviewer #2 Comments:**

In the manuscript "Assessment of a multi-resolution snow reanalysis framework: a multi-decadal reanalysis case over the Upper Yampa River Basin, Colorado", the authors present a multi-resolution approach for a snow data assimilation application in the mountainous region of the Colorado River Basin. They assimilate fSCA estimates from different satellite-based remote sensing systems and LSM results to generate a SWE reanalysis dataset. They present a comparison of prior and posterior (to the assimilation of remote sensed data) SWE results with station measurements as a validation of the assimilation framework. In the results section, the performance of the MR approach is discussed in detail. Main aim of the MR approach is reducing the computing time for large scale applications and datasets.

General Comments

The authors present a comprehensive application of a data assimilation and MR framework. They present state-of-the-art data and approaches and apply their own data assimilation and MR framework to a large test case. The approach to compare the MR case with a baseline resolution case in the results section is well chosen and nicely elaborated. However, the authors could think of shortening the results section at some points, because the results are somehow quite obvious

from the beginning of the results section on. The differences in posterior SWE when coarsening the resolution for less complex terrain in the MR method compared to the baseline case are very small or even negligible. This is an impressive result and underlines clearly the validity to use the MR approach when a reduction of CPU and storage resources is needed. However, as said, the results section could be much shorter, because the differences are so small that most of the figures present a more or less "perfect match". Besides showing the performance of the MR method, the 31-year dataset gives very interesting insights in the snow climatology of the region. The manuscript is technically of very high quality and well written, and I recommend its publication after minor revisions.

We welcome this suggestion to shorten the result section, and did so with an aim for improving its flow in the revised manuscript. Section 3.1 was shortened by removing the illustrative fSCA maps figure (formerly figure 5) and the focus was kept on the differences in fSCA distributions between the baseline and MR cases. Similarly, we removed the redundant table and figure of the ECDFs curves in section 3.2.3. Only Figure 10 was kept in the revised manuscript to showcase the good performance of the MR approach in the context of inter-annual variability. Finally, the analysis of the $SWE_{peak}$ standard deviation was removed from section 3.3, and we chose to focus on the coefficient of variation only. The scatter plot and difference distribution of the ensemble uncertainty coefficient of variation was added to Figure 11 based on Reviewer #1 comment (see above).

Specific Comments

L. 15: Rephrase and remove "by the user": e.g. "Factor 2 multiples . . . are chosen in this study as specific set of resolutions."

We agree with the reviewer, and modified the revised manuscript accordingly.

**List of all relevant changes made to the revised manuscript:**

1. Section 3.1 was shortened by removing the illustrative fSCA maps (formerly Figure 5)
2. Section 3.2.3 was shortened by removing the redundant analysis of the ECDFs curves (formerly Table 1 and Figure 12)
3. Section 3.3 now only focuses on the SWEpeak coefficient of variation, and the analysis of the standard deviation has been removed.
4. The scatter plot and difference distribution of the ensemble uncertainty coefficient of variation was added to Figure 11 in the revised manuscript

[revised manuscript text omitted]
  $<CV>$ (Figure 11) were created by pooling  $<CV>$ for each pixel from its mean and standard deviation over all 31 WYs as follows (Bingham and Fry (2010)):

$$\overline{<\sigma>_i} = \sqrt{\frac{\sum_{y=1}^{31}(<\sigma>_i^y)^2 + \sum_{y=1}^{31}(<\mu>_i^y - \overline{<\mu>_i})^2}{31}}$$

$$\overline{<CV>_i} = \frac{\overline{<\sigma>_i}}{\overline{<\mu>_i}} \tag{2}$$

15   where the overbar notation denotes the 31-year average. $\overline{<\sigma>_i}$ is the 31-yr average ensemble $SWE_{peak}$ standard deviation for pixel $i$, and $\overline{<\mu>_i}$ is the 31-year average ensemble $SWE_{peak}$ mean for the same pixel $i$. $<\sigma>_i^y$ and $<\mu>_i^y$ are respectively the ensemble $SWE_{peak}$ standard deviation and mean for each individual WY $y$. The 31-yr average $SWE_{peak}$ coefficient of variation ($\overline{<CV>_i}$) for each pixel $i$ was calculated as the ratio between the pixel 31-yr average ensemble $SWE_{peak}$ standard deviation and mean.

[Figure]

**Figure 11.** (a) Maps of  pixel-wise 31-year $SWE_{peak}$ coefficient of variation ($\overline{<CV>}$)  over the Upper Yampa River Basin for the 90 m baseline, the MR case, the percent difference between the two approaches (MR – baseline), and the corresponding scatter plot. (b)  Distribution of  $\overline{<CV>}$ relative difference with complexity metric (CM), elevation (Z),  slope, forested fraction (fVEG) , and  are shown in the right panel.  Pixels with a 31-yr average $SWE_{peak}$ lower than 5 cm were discarded from  the analysis.

As seen in Figure 11a, the spatial distributions of  $\overline{<CV>}$  is highly variable. For both the baseline and MR cases, the high elevation areas accumulating large amounts of SWE (see Figure 6a) show  a $\overline{<CV>}$ on the order of 10% – 20%, while the lower parts of the UYRB have a  $\overline{<CV>}$ higher than 60%. Regarding the relative difference between the MR and baseline cases (Figure 11

5   a), regions accumulating the most SWE with the lowest $\overline{<CV>}$ also have the lowest relative difference between the MR and baseline cases (white areas on the eastern and southern edges of the UYRB).  The basin-average difference in $\overline{<CV>}$ of 0.47% is

10  however negligible.

The spatial distributions of these differences are shown in Figure 11b. Besides highlighting again the low magnitude of the difference, its distribution is also in accordance with the way the MR was designed. Most of the difference observed in

$SWE_{peak}$ uncertainty is concentrated over areas of low complexity, elevation, slope, heterogeneously dense forests, and most importantly, low $SWE_{peak}$.

[revised manuscript text omitted]